# Developed meloxicam loaded microparticles for colon targeted delivery: Statistical optimization, physicochemical characterization, and *in-vivo* toxicity study

Syed Abdul Wasay[1], Syed Umer Jan[1]*, Muhammad Akhtar[2,3], Sobia Noreen[2], Rahman Gul[1]

1 Department of Pharmaceutics, Faculty of Pharmacy and Health Sciences, University of Balochistan, Quetta, Pakistan, 2 Department of Pharmaceutics, Faculty of Pharmacy, The Islamia University of Bahawalpur, Bahawalpur, Pakistan, 3 Department of Medical laboratory Technology, Faculty of Medicine and Allied Health Sciences, The Islamia University of Bahawalpur, Bahawalpur, Pakistan

* umer.pharm@um.uob.edu.pk

**Data Availability Statement:** All relevant data are within the manuscript.

## Abstract

The study aimed to fabricate and evaluate Meloxicam (MLX) loaded Hydroxypropyl Methylcellulose (HPMC) microparticles for colon targeting because MLX is a potent analgesic used in the treatment of pain and inflammation associated with colorectal cancer (CRC). Nevertheless, its efficiency is limited by poor solubility and gastrointestinal tracts (GIT) associated side effects. Seventeen formulations of MLX loaded HPMC microparticles were fabricated by the oil-in-oil (O/O)/ emulsion solvent evaporation (ESE) technique. A 3-factor, 3-level Box Behnken (BBD) statistical design was used to estimate the combined effects of the independent variables on the dependent variables (responses), such as the percent yield ($R_1$), the entrapment efficiency (EE) ($R_2$), mean particle size ($R_3$) and *in vitro* percentage of cumulative drug release ($R_4$). For physicochemical characterization FTIR, XRD, DSC, and SEM analyses were performed. Biocompatibility and non-toxicity were confirmed by *in-vivo* acute oral toxicity determination. The percentage yield and EE were 65.75–90.71%, and 70.62–88.37%, respectively. However, the mean particle size was 62.89–284.55 μm, and the *in vitro* cumulative drug release percentage was 74.25–92.64% for 24 hours. FTIR analysis showed that the composition of the particles was completely compatible, while XRD analysis confirmed the crystalline nature of the pure drug and its transition into an amorphous state after formulation. DSC analysis revealed the thermal stability of the formulations. The SEM analysis showed dense spherical particles. The toxicity study in albino rabbits showed no toxicity and was found biocompatible. The histopathological evaluation showed no signs of altered patterns. Results of this study highlighted a standard colonic drug delivery system with the ability to improve patient adherence and reduce GIT drug-associated side effects in CRC treatment.

**Funding:** The author(s) received no specific funding for this work.

**Competing interests:** The authors have declared that no competing interests exist.

## 1. Introduction

Colorectal cancer (CRC) is a chronic heterogeneous disease caused by genetic mutation wherein various pathways could have participated in tumor commencement, evolution, and growth [1, 2]. Inflammatory-motivated genetic change, and epigenetic modification, are substantial features of CRC tumor commencement [3]. Inflammation is a rapid biochemical response added on through prostaglandin E2 (PGE2), the primary inputs of this response are corrosive chemicals, antigen-antibody reactions, and mechanical trauma [4]. In the CRC growth, various cytokines like TNF-α, IL-1, and IL-6 are involved. TNF-α is released by macrophages or monocytes which upholds the development of tumor growth, angiogenesis, and long-lasting inflammation, and likewise, IL-1 galvanizes pro-inflammatory and up-regulatory responses [5]. Surgery, radiation, and chemotherapy are the most common traditional treatment approaches for many types of cancers including CRC. More than 40% of cancers are treated with surgery (full removal of the tumor); as a result, surgery is a common treatment choice for a variety of cancer, whether complete or partial excision is used [6]. Radiation therapy, whether used alone or in combination with surgery or chemotherapy, is one of the most used approaches for treating cancer. However, this approach is frequently connected with the interaction of radiations with DNA, followed by the creation of a free radical [7]. Chemotherapeutic agents are another traditional treatment approach for treating a range of cancers. Though the use of chemotherapeutic drugs for cancer treatment and/or improving the patient's quality of life is almost complementary, the major issues concerning chemotherapy must be addressed, which include low bioavailability due to poor blood flow, inability to reach the target site due to interaction with the reticuloendothelial system (RES), and lack of tumor specific targeting [8]. The clinical importance of traditional approaches (i.e., surgery, radiation, and chemotherapy) is restricted due to these limitations. In 2018, more than 1.8 million cases were detected and CRC was rated the third most common cancer. In terms of mortality, 881,000 deaths were associated with CRC and categorized it as the second of all diseases [9].

Meloxicam (MLX), 4-Hydroxy-2-methyl-N-(5-methyl-2-thiazolyl)-2H-1,2- benzothiazine-3-carboxamide 1,1-dioxide pertains to the enolic acid group of oxicam derivatives [10]. In the solubility and permeability profile, MLX is categorized as class-II of the BCS system of classification [11]. It exhibits poor water solubility and a low dissolution rate (almost 4.4 μg/mL at water), besides an elimination half-life of approximately 20 hrs [12]. It is commonly used for the management of acute pain, inflammation, and stiffness induced by rheumatoid arthritis, ankylosing spondylitis. osteoarthritis, injuries, and tendinitis [13]. Many population-based retrospective and prospective studies have found that regular usage of selective cyclooxygenase-2 (COX-2) inhibitors such as MLX are connected to a lower incidence and mortality rate of CRC [14]. MLX's limited solubility causes poor dissolution and low absorption from the gastrointestinal tract (GIT) at physiologic pH, limiting its therapeutic efficacy [15]. The gastrointestinal side effects of MLX such as dyspepsia, ulceration, bellyache, and bleeding significantly limit its clinical application which may also restrict its long-term usage for CRC prevention [16]. As a result, developing a suitable drug carrier system for efficient and controlled delivery of MLX to the colonic region is vital.

Controlling the drug release is critical for optimal delivery of the medicine at the site of action after oral administration. A controlled release delivery system has the capability to maintain a consistent plasma drug concentration for an extended period, reducing the adverse effects associated with traditional dose forms [17]. Unfortunately, poor drug solubility, degradation, low bioavailability, and bio-distribution make it difficult to pinpoint the site of action [18]. Encapsulating the drug in a polymeric matrix that allows for precise and controlled drug release at a steady rate for a long period is one strategy to address low solubility and poor bioavailability [19]. Polymeric particulate systems, such as microparticles, nanoparticles, and

microsponges have gotten a lot of attention in recent years due to their various and customizable features [20]. Meanwhile, microparticles and nanoparticles are commonly prepared with biocompatible and biodegradable polymers, as drug carriers to overcome the low solubility, limited bioavailability, drug degradation, and to manage controlled released delivery at the site of action [21]. Microparticles with diameters ranging between 1–1000 μm are spherical particles with an active pharmacological ingredient in the core and a polymeric coating that normally controls drug release from the microparticles [18] and can be fabricated by numerous methods such as solvent evaporation, fluidized bed method, conservation method, spray drying, and interfacial polymerization method [22, 23]. In the present research work, microparticles were fabricated through the oil in oil (O/O) emulsion solvent evaporation (ESE) method, because it is simple to make, does not require harsh processing conditions, and also, does not impact drug activity [24, 25]. It is mostly used to microencapsulate drugs that dissolve in the dispersion phase and have low aqueous solubility [26].

HPMC is a semi-synthetic ether derivative of cellulose that is frequently used in a variety of fields, such as pharmaceutical, drug delivery, and food industry as a stabilizer, thickener, and emulsifier [27]. Also, it is widely used in the development of controlled-release devices, because of its non-toxic properties and ease of production [28]. HPMC is a member of the swellable hydrophilic medium systems, when exposed to aqueous solutions, it produces a gel layer which is a promising factor in controlled release patterns. Drug release from HPMC matrices has been reported to be affected by: (a) the polymer physical properties, for example, drug/polymer ratio, polymer viscosity, and particle size; (b) the drug physicochemical properties, for example, solubility, and particle size, and, (c) fabricating factors, for example, stirring speed, excipients of formulation, and processing techniques [29]. Mohammad El-Badry [30] prepared HPMC microparticles by freeze-drying technique, using Albendazole as the model drug.

Response surface methodology (RSM) is a set of mathematical and statistical processes for analyzing and optimizing the effects of independent variables on dependent variables using the design of experiments. In comparison to other RSM designs, the Box-Behnken design (BBD) was chosen for the optimization of dependent variables since it requires fewer trial combinations, is efficient, cost-effective, and takes less time [31]. Regression equations, often known as models, are used to represent the answers quantitatively. Furthermore, this technique has advantages over traditional optimization methods, which are costly, time-taking, and require a significant number of reagents for trials [32].

The goal of the designed study was to formulate MLX loaded HPMC microparticles by using a 3-factor, 3-level Box Behnken design through the oil in oil (O/O) ESE technique for colon targeted delivery. Out of the 17 fabricated microparticle formulations, one was statistically optimized based on the percentage yield, EE, particle size, and in-vitro cumulative drug release, and thereby further evaluated oral toxicity studies for biocompatibility and non-toxicity confirmation in terms of clinical, biochemical, and histopathological markers, using rabbit as an animal model. To the best of our knowledge, this is the first systematic investigation using a statistical design to report the utilization of MLX as a model drug and HPMC polymer as a matrix component for microparticle development through the oil in oil (O/O) ESE technique for colon targeted delivery. Furthermore, the influence of three independent variables on four dependent variables was studied.

## 2. Materials and methods

### 2.1. Materials

Meloxicam and HPMC were donated for the research work by English Pharmaceutical Industries, Lahore (Pakistan), and Martin Dow Marker Limited (formerly MERCK Pvt Ltd), Quetta

(Pakistan), respectively. Hydrochloric acid (37%) and n-Hexane were procured from AnalaR BDH Laboratory (UK). Sodium hydroxide, and ethanol, were procured from Evonik Roehm GmbH (Germany). Dichloromethane and Liquid paraffin were purchased from Merck KGaA (Germany), while Span-80 was acquired from Avonchem Ltd (UK). All chemicals of analytical grade were used for the study.

## 2.2. Method

**2.2.1. Experimental design (Box–Behnken design).** As a statistical tool and mathematical approach, RSM was used to study the impact of various formulation variables on the microencapsulation process. With the help of a 3-factor,3-level BBD seventeen experimental runs were designed [25]. To obtain optimized MLX loading HPMC microparticles, the three most influential independent process variables (factors) were selected based on their compact influence on physicochemical properties of microparticles, i.e., the drug to polymer ratio ($X_1$), the stirring speed ($X_2$), and the concentration of surfactant ($X_3$), to each one with three levels: [$X_1$ (1:1, 1:2.5, and 1:4 mg), $X_2$ (800, 1000, and 1200 rpm), and $X_3$ (0.5, 1, and 1.5%)] **Table 1**.

The impact of modification in independent variables on the dependent variables such as percentage yield ($R_1$), EE ($R_2$), average particle size ($R_3$), and *in vitro* percentage of cumulative drug release ($R_4$) was evaluated by employing a one-way analysis of variance (ANOVA) practicing Stat-Ease Design-Expert$^{\circledR}$ (Design Expert 11.1.2.0 x 64) software with expanded capabilities for data analysis [33]. The significance for each independent variable was evaluated by

**Table 1. Box Behnken Design (BBD) based experimental parameters.**

| Independent variables | Levels | | |
|---|---|---|---|
| | Low | Medium | High |
| $X_1$ = Drug to polymer ratio (mg) | 1:1 | 1:2.5 | 1:4 |
| $X_2$ = Stirring speed (rpm) | 800 | 1000 | 1200 |
| $X_3$ = Surfactant concentration (%) | 0.5 | 1 | 1.5 |

| Dependent variables | Code | Drug (MLX)(mg) | $X_1$ (mg) | $X_2$ (rpm) | $X_3$ (%) |
|---|---|---|---|---|---|
| $R_1$ = Percentage yield (%), | F1 | 200.00 | 500 | 1000 | 1.0 |
| $R_2$ = Entrapment efficiency (EE) (%), | F2 | 200.00 | 500 | 1200 | 1.5 |
| $R_3$ = Mean particle size (μm), and | F3 | 200.00 | 200 | 1200 | 1.0 |
| $R_4$ = *In vitro* percentage of cumulative drug release (%). | F4 | 200.00 | 500 | 1000 | 1.0 |
| | F5 | 200.00 | 800 | 1000 | 0.5 |
| | F6 | 200.00 | 800 | 800 | 1.0 |
| | F7 | 200.00 | 500 | 1000 | 1.0 |
| | F8 | 200.00 | 500 | 1200 | 0.5 |
| | F9 | 200.00 | 500 | 800 | 1.5 |
| | F10 | 200.00 | 800 | 1200 | 1.0 |
| | F11 | 200.00 | 200 | 1000 | 0.5 |
| | F12 | 200.00 | 800 | 1000 | 1.5 |
| | F13 | 200.00 | 200 | 1000 | 1.5 |
| | F14 | 200.00 | 500 | 1000 | 0.5 |
| | F15 | 200.00 | 500 | 1000 | 1.0 |
| | F16 | 200.00 | 500 | 1000 | 1.0 |
| | F17 | 200.00 | 200 | 800 | 1.0 |
| | F0 | 200.00 | 500 | 1000 | 1.0 |

using the following non-linear quadratic expression (1).

$$Y = b_0 + b_1X_1 + b_2X_2 + b_3X_3 + b_{12}X_1X_2 + b_{13}X_1X_3 + b_{23}X_2X_3 + b_{11}X^2_1 + b_{22}X^2_2 + b_{33}X^2_3 \qquad (1)$$

Where:

Y is the dependent variable,

$X_1$, $X_2$, $X_3$ are independent variables,

$b_0$ as an intercept,

$b_1$, $b_2$, $b_3$ are non-linear coefficients,

$b_{11}$, $b_{22}$, $b_{33}$ are squared coefficients, and

$b_{12}$, $b_{13}$, $b_{23}$ are interaction coefficients of this quadratic equation.

**2.2.2. Microparticles fabrication.** The MLX loaded HPMC microparticles were fabricated by the oil in oil (O/O) / ESE method [26, 34]. The design determined amount of HPMC was dissolved bit by bit in the ethanol-dichloromethane solution, ratio (1:1) over the magnetized stirrer (Velp Scientifica, Usmate (MB), Italy) at 250 rpm. Consequently, the exact quantity of MLX was dispersed within the polymeric solution. In the internal phase, magnesium stearate (100mg) was added as a drop stabilizer with incessant stirring. The external phase was prepared in a 250 ml beaker by the addition of 50 ml of liquid paraffine and designed determined concentration of span-80 was added as surfactant. Eventually, the external phase was added to the internal phase dropwise with care and continuously stirred by a tri-blade propeller (Eurostar IKA, WERKE), with a designed proposed stirring speed for 3–4 hr. or till perfect evaporation of the organic solvent. The fabricated microparticles were decanted and filtered with the aid of Whatman No.42 filter paper and then washed 4–5 times with n-hexane (40 ml) for absolute removal of liquid paraffin and dried at ambient temperature for 24 hrs [25].

## 2.3. Characterization of microparticles

**2.3.1. Determination of percentage yield.** The percentage yield of completely dehydrated microparticles was computed by the actual yield divided by the theoretical yield as well as multiplying the received ratio by 100 [35]. The percentage yield was calculated by Eq (2), as under:

$$Percentage\ Yield = \frac{Practical\ Yield}{Theoritical\ Yield} \times 100 \qquad (2)$$

**2.3.2. Entrapment efficiency (EE).** Accurately weighed 50mg of fabricated microparticles were taken and blended in a mortar followed by dispersion in 50 ml of dichloromethane. The dispersion was applied for 2 hrs stirring, and further sonicated for half-hour to completely remove the drug from the blended powder in the extracting medium. After purification with the aid of Whatman No.42 filter paper, the filtrate was further diluted in Phosphate buffer (pH 6.8) and taken three times absorbance at $\lambda_{max}$ 362nm to calculate the concentration employing UV Spectrophotometer (IRMECO Gmbh, Gaeltacht, Germany) [36]. The percentage of EE was determined by Eq (3), as under:

$$E.E\ (\%) = \frac{Actual\ amount\ of\ MLX\ in\ microspheres}{Theoretical\ amount\ of\ MLX\ in\ microspheres} \times 100 \qquad (3)$$

**2.3.3. Fourier Transform Infrared Spectroscopic (FTIR) analysis.** To determine the compatibility between MLX, polymer (HPMC), physical mixture of drug and polymer, and MLX loaded optimized formulation (F0), an FTIR spectrophotometer (Bruker, Tensor 27,

Germany) was used in the scanning range of 4000-400$^{cm^{-1}}$. The analyzing sample was located at the disc and the plunger was properly constrained by rotating the arm for enough touch with the sample and scanning turned into completed in 16 sec [37].

**2.3.4. Determination of particle size.** All formulation's volumetric type average particle sizes were determined by the most frequently applied method of optical microscopy, an optical microscope (Eclipse E-200 LED, Nikon, Tokyo, Japan) was used. Before starting the study, the eyepiece micrometer was standardized by employing the stage micrometer. To put together a slide for analysis, a small number of microparticles were located on the slide and watched beneath the lens at 10X. With the assistance of an eyepiece micrometer, the particle size of various microparticles was measured [35]. The mean particle size of all formulations was calculated by Eq (4), as under:

$$Mean\ particle\ size = \frac{Sum\ of\ diameter\ of\ obsereved\ particles}{Number\ of\ observed\ particles} \tag{4}$$

**2.3.5. Thermal stability via Differential Scanning Calorimetric (DSC) analysis.** Differential scanning calorimetry (DSC) of MLX, polymer (HPMC), Physical mixture of drug and polymer, and MLX loaded optimized formulation (F0) microparticles was performed using DSC (LAB KITS-100, Hong Kong). Test sample $7 \pm 0.1$ mg was deposited on an aluminum pan heated to 30–300°C at a flow rate of 20 mL/min. Nitrogen was used as a purge gas, while indium and zinc were used as standard [38, 39].

**2.3.6. X-Ray Diffraction (XRD) analysis.** Diffraction trends of MLX, polymer (HPMC), physical mixture of drug and polymer, and MLX loaded optimized formulation (F0) microparticles were examined using X-ray diffractometry (JEOL, JDX-3532, Japan) under 30 mA and 35 kV working conditions. The samples were examined at a rate of 2θ/min in the 5–70 range. The gained results were analyzed and compared for the existence of peaks, their location, and shifting [35].

**2.3.7. Scanning Electron Microscopic (SEM) analysis.** The scanning electron microscope (JSM5910, JEOL, Tokyo, Japan) was used to examine the surface morphology and shape of the MLX loaded optimized formulation (F0). The sample was put on metal stubs using dual-sided adhesive tape for SEM measurements. It was dried in a vacuum chamber before being sputter coated with a gold coating and examined using a high-resolution scanning electron microscope at various magnifications [38].

**2.3.8. *In vitro* study of drug release.** The MLX loaded HPMC microparticles *in vitro* drug release behavior was determined by employing a USP dissolution equipment type-II (Pharma test Hainburg, Germany) at 50 rpm, 37 °C ± 0.5 temperature. Dissolution mediums, pH 1.2, 6.8, and 7.4, were used successively for 2, 10, and 12 hours, respectively, in a sequential pH change approach [40]. A precisely weighed quantity of samples equivalent to 7.5mg of MLX was transferred to a dialysis membrane (12-14KDa) (Medicell Membrane Ltd, UK). It had earlier been soaked in release media for almost 12 hrs. Two clamps were used to secure the dialysis membrane's open ends and immersed in 450 ml of simulated gastric medium pH 1.2 for 2 hrs. After that, the simulated gastric medium pH 1.2 was changed with Phosphate buffer 6.8 for 10 hrs., and finally, it was changed with Phosphate buffer 7.4 for a further 12 hrs. The 5ml of dissolution medium was pulled out at prior set time intervals of 0.25, 0.5, 1, 2, 4, 6, 8, 10, 12, 16, 20, and 24 hrs, followed by the addition of an equal volume of a fresh dissolution medium to uphold the required sink conditions throughout the analysis [39]. Filter the pulled-out medium and take the three-time absorbance of the filtrate by employing a UV-Spectrophotometer (IRMECO Gmbh, Gaeltacht, Germany) at λ$_{max}$ 362 nm. With the aid of a regression

mathematical equation, the concentration of MLX was computed employing a calibration curve. The percentage of cumulative drug release was calculated by Eq (5), as under:

$$Percentage\ of\ drug\ release = \frac{amount\ of\ the\ drug\ release\ at\ time\ (t)}{amount\ of\ the\ drug\ entraped\ in\ microparticales} \times 100 \quad (5)$$

**2.3.9. Kinetic models trends of *in vitro* drug release.** Various kinetic models were applied to *in vitro* drug release data to determine the order and mechanism of drug release from the formulations. The *in vitro* drug release data were subjected to regression analysis, using a coefficient of zero-order as the cumulative quantity of drug release vs. time [41]. First-order as the log cumulative release of drug vs. time [42], Higuchi as the cumulative quantity of drug released vs. square root of time [43], and Korsmeyer Peppas models [44]. The correlation coefficient ($R^2$) for the different kinetic models and the diffusion exponent (n) values for the Korsmeyer-Peppas models were determined by the DDsolver.xla. If the value of "n" is 0.5, the preparation is Fickian diffusion; if the value of "n" is greater than 0.5 but less than 1.0 (0.5 < n < 1.0), the release is non-Fickian diffusion (anomalous diffusion). If "n" is 1.0, the preparation complies with case -II transport; if it is greater than 1.0, the release follows super case-II transport [45].

**2.3.10. Acute oral toxicity study.** The acute oral toxicity study of MLX-loaded HPMC microparticles was performed to investigate the safety and biocompatibility of microparticles on albino rabbits, following the principles of the Organization for Economic Cooperation and Development (OECD) [46]. Albino rabbits were chosen as an animal model for the study because of their well-established pathophysiology and accessible data from which to judge the effects on human wellbeing [47]. The Institutional Animal Ethical Committee (IAEC), The Faculty of Pharmacy and Health Science, University of Balochistan (UOB), Quetta, Pakistan, reviewed and approved the study protocols (Ref letter NO. FoP & HS/ICE/212/20, dated 20-11-2020). Twelve male rabbits weighing between 1.70–2.5 kg were marked to allow the individual recognition and split into two groups (n = 6), each labeled as group-I (control) and group-II (test). The rabbits were housed alone in cages that were cleaned and ventilated, with access to food and water. The animal transitory room conditions were maintained following OECD norms, i.e., ambient temperature (25°C ± 2), relative humidity (40%), and the artificial lighting was kept on for 12 hours of brightness, and 12 hours of darkness. The rabbits were fasted all night apart from the water before drug therapy. The group-I (control) was given no drug treatment but water and food, while the group-II (test) was given optimized MLX loaded microparticles equivalent to the drug 1.5 mg/kg body weight by oral gavage a flexible feeding tube of 20 gauge. This study was carried out for 14 days and the rabbits were observed for food and water intake, body weight, the sign of illness, any kind of seeable skin irritation/toxicity, and mortality. A possible source of suffering in animal research is pain induced by experimental procedures, injuries, and diseases [48]. To alleviate the stress/suffering of rabbits, a parenteral anesthetic combination like ketamine and xylazine have become the agents of choice for rabbit anesthesia due to their efficacy, low cost, and ease of administration [49]. The rabbits were anesthetized with a combination of ketamine (35 mg/kg) and xylazine (5 mg/kg) on the 15th day of the experiment. The drugs were mixed in a single syringe, swabbed the area with 70% of ethanol and injected intramuscularly into the quadriceps femoris muscles while the rabbits were sternally recumbent in the table [50]. The blood samples were obtained instantly in ethylenediaminetetraacetic acid (EDTA) tubes to avoid blood coagulation from both groups for blood biochemistry. The rabbits were subsequently sacrificed under anesthesia by decapitation using a pair of shear/scissor blades, cut between the base of the head and the top of the

neck in one swift, smooth motion. The key organs were removed, weighed, and kept separately in 10% formalin solution for histopathological studies [33, 51, 52].

# 3. Results and discussion

## 3.1. MLX loaded HPMC microparticles

Light yellow colored and spherical shaped MLX loaded HPMC microparticles were successfully fabricated for colon targeted drug delivery by the oil in oil (O/O) ESE technique, using 3-factor-3-level statistical design. With the help of design, one formulation was taken as an optimized formulation based on percentage yield, EE, particle size analysis, and cumulative percentage drug release study. Physicochemical characterizations and oral toxicity studies were carried out to determine the microparticles components compatibility and biocompatibility, respectively.

## 3.2. Box Behnken design

**3.2.1. Percentage yield.** ]The percentage yield ($R_1$) ranged from 65.75–91.71%, with F6 (91.71 ±2.65) having the highest percentage yield and F11 (65.75 ±1.31) having the lowest, as shown in Table 2.

The quadratic equation for % yield with the independent variable is as under in Eq (6)

$$\begin{aligned} \text{Percentage yield } (R_1) \\ = +80.50 + 8.32A - 3.81B - 1.68C - 2.31AB - 2.73AC + 0.9925BC + 2.07A^2 \\ - 1.17B^2 - 2.78C^2. \end{aligned} \tag{6}$$

Due to the increased viscosity, thickness, and reduced syringeability of the polymeric solution, the percentage yield increases significantly ($p < 0.05$) with increasing polymer content, as depicted in Fig 1(A) and 1(B) [35]. The percent yield improved when the content of the

**Table 2. Observed values of independent variables in BBD.**

| Code | Percentage yield (%) | Entrapment efficiency (%) | Particle size (μm) | Drug release (%) |
|---|---|---|---|---|
| | ($R_1$) | ($R_2$) | ($R_3$) | ($R_4$) |
| F1 | 76.87 ± 1.00 | 82.5 ± 0.97 | 151.79 ± 20.88 | 86.44 ± 1.94 |
| F2 | 70.98 ± 3.18 | 86.5 ± 2.14 | 81.20 ± 8.97 | 92.64 ± 0.81 |
| F3 | 68.44 ± 0.79 | 79.16 ± 0.97 | 62.89 ± 12.35 | 83.58 ± 1.49 |
| F4 | 81.59 ± 1.18 | 85.32 ± 1.12 | 182.29 ± 25.58 | 82.03 ± 1.75 |
| F5 | 88.33 ± 1.24 | 83.41 ± 0.94 | 228.79 ± 18.83 | 83.83 ± 1.84 |
| F6 | 90.71 ± 2.65 | 85.95 ± 0.57 | 276.23 ± 13.99 | 74.25 ± 1.28 |
| F7 | 80.39 ± 0.95 | 81.9 ± 2.27 | 152.65 ± 34.53 | 84.76 ± 1.70 |
| F8 | 72.97 ± 1.43 | 74.45 ± 2.82 | 106.96 ± 11.28 | 87.72 ± 0.66 |
| F9 | 78.15 ± 1.30 | 87.55 ± 1.29 | 102.21 ± 16.91 | 86.48 ± 1.23 |
| F10 | 79.99 ± 1.11 | 83.51 ± 2.52 | 203.53 ± 14.10 | 84.51 ± 1.46 |
| F11 | 65.75 ± 1.31 | 70.62 ± 1.38 | 70.34 ± 10.57 | 87.49 ± 1.87 |
| F12 | 80.11 ± 1.78 | 88.37 ± 1.64 | 128.12 ± 17.77 | 86.7 ± 1.22 |
| F13 | 68.44 ± 1.71 | 88.33 ± 1.63 | 107.18 ± 17.71 | 89.32 ± 2.06 |
| F14 | 84.11 ± 0.46 | 74.75 ± 1.74 | 284.55 ± 9.35 | 84.21 ± 1.01 |
| F15 | 82.25 ± 2.76 | 79.58 ± 1.55 | 150.54 ± 18.17 | 85.12 ± 1.46 |
| F16 | 81.4 ± 1.51 | 83.78 ± 1.10 | 165.36 ± 11.19 | 85.41 ±1.93 |
| F17 | 77.91 ± 0.96 | 77.48 ± 1.61 | 98.46 ± 14.62 | 84.24 ± 1.91 |

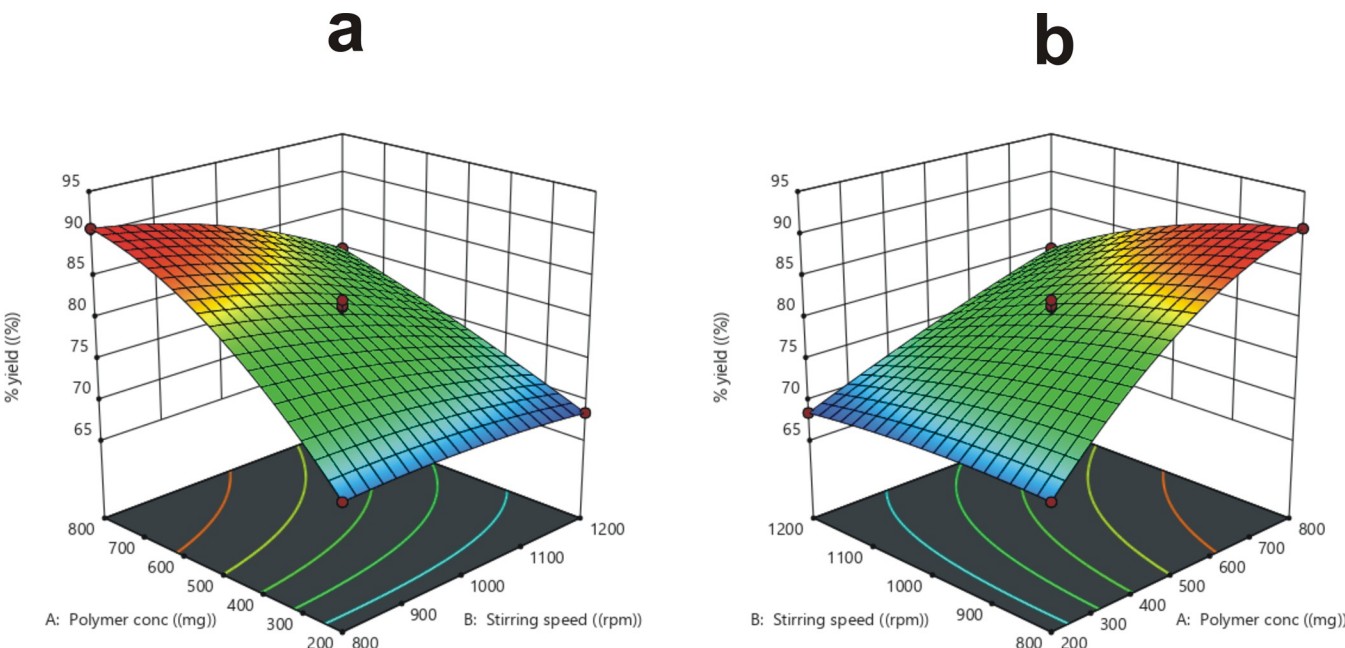

**Fig 1. The influence of polymer concentration and stirring speed on percentage yield (a), the influence of surfactant conc and polymer conc on percentage yield (b).**

surfactant was increased from 0.5 to 1.5% due to particles alignment. The percentage yield was reduced with increased stirrer speed because of foaming formation, turmoil, and sticking of microparticles with the container walls. Additionally, during the washing process, microparticles waste also reduced the percentage yield [25].

**3.2.2. Entrapment efficiency (EE).** At a constant drug concentration, the influence of polymer concentration ($X_1$), stirring speed ($X_2$), and surfactant content ($X_3$) on the entrapment efficiency (EE) ($R_2$) of microparticles was investigated, and shown in Fig 2(C) and 2(D). F12 (polymer concentration: 800 mg, stirring speed: 1000 rpm, surfactant concentration: 1.5%) depicted the highest percentage of EE 88.37% ±1.64, while F11 (polymer concentration: 200 mg, stirring speed: 1000 rpm, surfactant concentration: 0.5%) the lowest percentage of EE 70.62% ±1.38, as revealed in Table 2. According to ANOVA in Table 3, the change in polymer concentration and surfactant content had a significant ($p < 0.05$) effect on the EE, whereas the stirring speed had an insignificant ($p > 0.05$) effect on it.

The quadratic expression for entrapment efficiency (EE) with the independent variable is as under in Eq (7).

$$EE\% \ (R_2) = +82.62 + 4.24A - 0.2637B + 4.90C - 1.03AB - 1.26AC - 0.187BC \\ - 0.5730A^2 - 0.5180B^2 - 1.29C^2. \tag{7}$$

The influence of polymer concentration on EE showed a significant ($p < 0.05$) increase. High polymer concentration generates a condensed structure due to the increased amount of accessible polymer in the internal phase and non-slenderness of the polymer network, which minimized drug loss to the external phase [39]. An increase in stirring speed from 800–1200 rpm resulted in a non-significant ($p > 0.05$) reduction in EE. This is attributed to a reduction in particle size, which increases surface area and thus decreases drug diffusion into the external phase [53, 54]. It was also observed that when the surfactant content was enhanced from 0.5–1.5%, the EE was significantly ($p < 0.05$) increased owing to the small size of the droplet

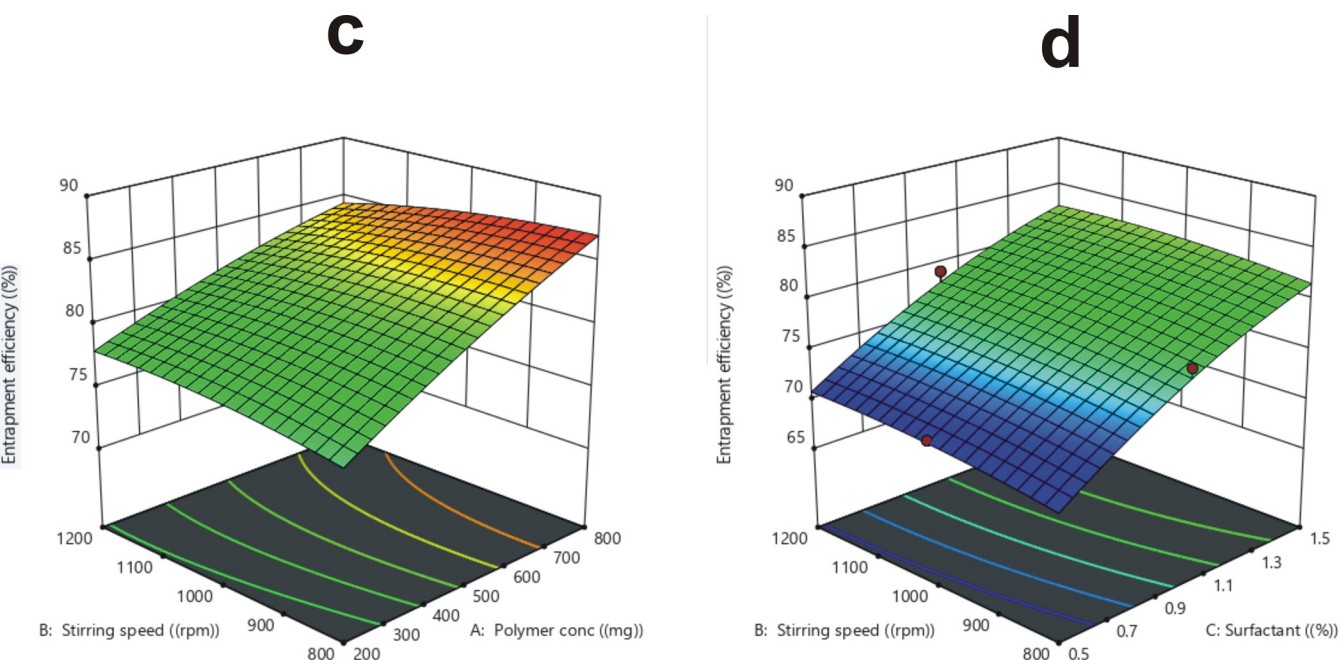

**Fig 2. The influence of stirring speed and polymer concentration on EE (c), the influence of stirring speed and surfactant concentration on EE (d).**

during the microparticles fabrication process [55]. Aside from that, the emulsifying agent forms a protective layer across the droplets, preventing them from coalescing [56].

**3.2.3. Particle's size.** The polymer concentration ($X_1$), stirring speed ($X_2$), and surfactant content ($X_3$), all influence the mean particle size ($R_3$) of MLX loaded HPMC microparticles. The F14 (polymer concentration: 500 mg, stirring speed: 1000 rpm, surfactant concentration: 0.5%) have the biggest particle size of 284.55 ± 9.35μm, while F3 (polymer concentration: 200 mg, stirring speed: 1200 rpm, surfactant concentration: 1.0%) have the smallest particle size of 62.89 ±12.35μm, as revealed in Table 2. The independent variable influences, such as polymer content, stirring speed, and surfactant concentration interaction was established by ANOVA as shown in Table 3.

**Table 3. ANOVA analysis values of independent variables.**

| Independent variables (Responses) | Statistical terms | *P-value* |
|---|---|---|
| Percentage yield (%) ($R_1$) | Polymer content ($X_1$) | < 0.0001 |
| | Stirring speed ($X_2$) | 0.0007 |
| | Conc of surfactant ($X_3$) | 0.0371 |
| Entrapment efficiency (%) ($R_2$) | Polymer content ($X_1$) | 0.0016 |
| | Stirring speed ($X_2$) | 0.7664 |
| | Conc of surfactant ($X_3$) | 0.0007 |
| Particle size (μm) ($R_3$) | Polymer content ($X_1$) | 0.0007 |
| | Stirring speed ($X_2$) | 0.0089 |
| | Conc of surfactant ($X_3$) | 0.0191 |
| Drug release (%) ($R_4$) | Polymer content ($X_1$) | 0.0012 |
| | Stirring speed ($X_2$) | 0.0009 |
| | Conc of surfactant ($X_3$) | 0.0039 |

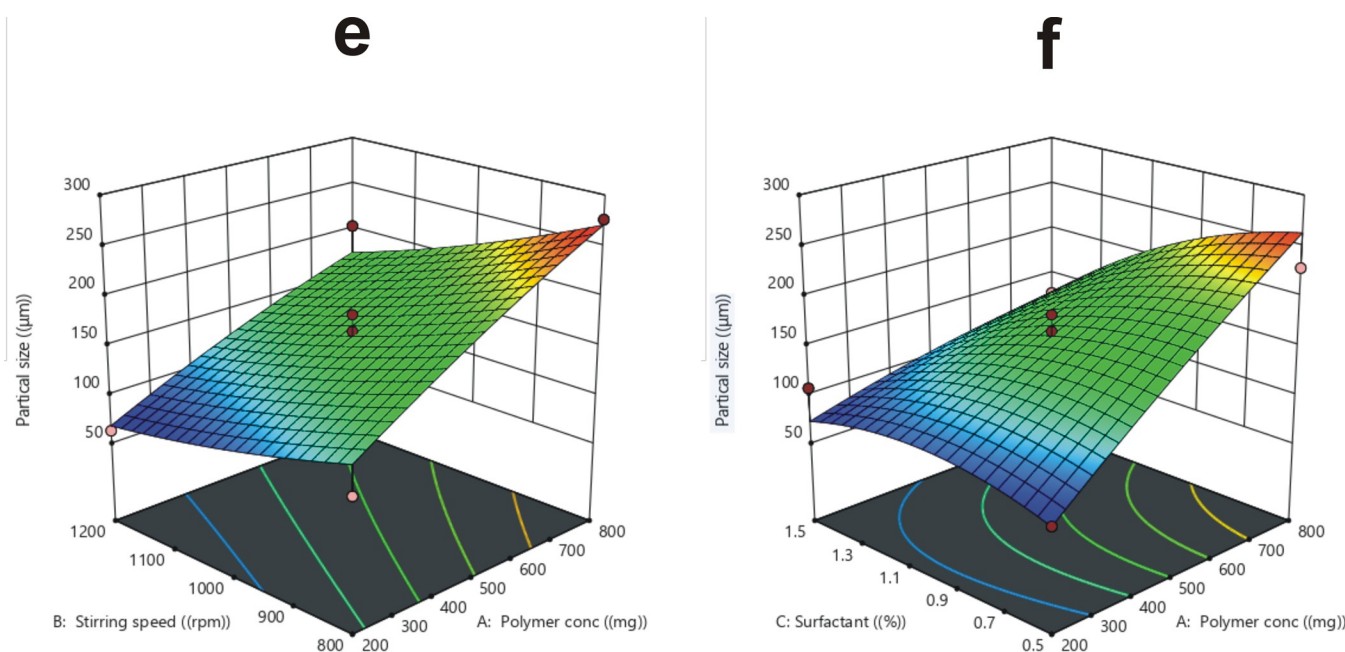

**Fig 3. The influence of stirring speed and polymer concentration on particles size (e), the influence of surfactant conc and polymer concentration on particles size (f).**

For mean particles size, quadratic equation along with independent variable is asunder in Eq (8).

$$\text{Mean particles size } (R_3) = +160.53 + 62.22A - 39.24B - 33.12C - 9.29AB - 34.38AC - 37.40BC - 6.60A^2 + 5.81B^2 - 20.86C^2. \tag{8}$$

When the concentration of polymer was increased from 200–800 mg, it depicted a significant ($p < 0.05$) impact on average particle size (R3), as shown in Fig 3(E) and 3(F). The generation of viscosity and thickness in the emulsion is the primary cause of mean particle size enlargement, which impedes dispersion and leads to the formation of bigger globules [57, 58]. The particle size decreased significantly ($p < 0.05$) when the stirrer speed was increased from 800–1200 rpm, which is attributed to an increase in circulating force due to high stirring speed, which reduces droplet size in the emulsion, therefore reducing the mean particle size as well [59]. A significant ($p < 0.05$) reduction in particle size was observed when surfactant content was increased from 0.5–1.5%, owing to a sufficient increase in the interfacial force of emulsion droplets, resulting in improved coalescency, which leads to the fabrication of smaller sized microparticles [39, 60].

**3.2.4. In vitro drug release.** The ability of drug molecules carried by polymer to reach the active site in sufficient quantities is a critical aspect of effective drug delivery. For this aim, while designing microparticulate drug delivery systems, we must investigate factors of drug release performance and polymer degradation. The following factors influence the rate of drug release from microparticles:

i. Solubility

ii. Drug diffusion from polymer

iii. Diffusion of surface-bound and adsorbed drug

iv. Erosion followed by degradation of the matrix of microparticles [47]

The *in vitro* drug release of MLX loaded HPMC microparticles were used to simulate *in vivo* release behavior [61]. The percentage of cumulative drug release of all 17 formulations (F1-F17) is given in Table 2. The F2, F13, and F8 depicted a higher percentage of cumulative drug release, while F6 had the lowest percentage.

The following is the percentage of cumulative drug release from all 17 formulations, in descending order:

F2 > F13 > F8 > F11 > F12 > F9 > F1 > F16 > F15 > F7 > F10 > F17 > F14 > F3 > F4 > F5 > F6.

For *in vitro* drug release the quadratic equation is as under in Eq (9)

$$\text{In vitro drug release } (R_4) \\ = 84.75 - 2.29A + 2.41B + 1.86C + 2.73AB + 1.01AC + 0.6600BC - 2.39A^2 \\ - 0.7160B^2 - 3.721C^2. \tag{9}$$

Fig 4(A)–4(C) demonstrates the in-vitro drug release profile of all formulations (F1-F17) employing various buffers (pH 1.2, 6.8, and 7.4). In the acidic medium of 0.1 N HCl at pH 1.2, the MLX loaded HPMC microparticles released nearly 1–8% of the drug, which is within the United States Pharmacopoeia 24 (USP 24) limit. According to this, in the acidic environment of the stomach, an enteric-coated formulation should not release more than 10% of the drug in 2 hrs, and this could be linked to the presence of solid drug crystals on microparticle surfaces [25]. The drug release increased after two hours when the acidic medium was changed to a basic medium of phosphate buffer pH 6.8 for 10 hrs., followed by phosphate buffer pH 7.4 for a further 12 hrs. So, for all formulations, the drug release trend was persisted in ascending order, the reason behind this behavior is the HPMC release mechanism, which includes wetting, hydration, swelling, and gel layer formation. This demeanor functions as a drug release barrier reliant upon the rate of gel layer interruption, drug diffusion rate, and corrosion of the system [62, 63].

As the polymer matrix broadness increased with the increase in polymer concentration from 200 to 800 mg, drug release ($R_4$) from microparticles was significantly ($p < 0.05$) prolonged, because the drug had to pass through an elongated dispersion pathway. Furthermore, increasing polymer concentration may result in larger particle size and reduced surface area [64, 65]. As indicated in Fig 5(G) and 5(H), the drug release was more rapid and significant ($p < 0.05$) from microparticles generated at higher stirring speeds, spanning from 800–1200 rpm, due to smaller particle size, large surface area, and fast wettability of microparticles with GIT fluent [66]. Higher surfactant content also caused a significant ($p < 0.05$) increase in drug release, which may be attributed to the particle size and surface area relationship, as well as excess availability of the free drug at the microparticles' surface [67, 68].

**3.2.5. Kinetics modeling data of drug release.** To anticipate the order and mechanism of drug release from MLX loaded HPMC microparticles, different kinetic models were used for the drug release data. These kinetic models' overall results are decisive in determining the most appropriate formulation. $R^2$ values are used to determine the best release strategy. When *in vitro* release data from all formulations was fitted to release kinetic modeling, the correlation coefficients ($R^2$) were computed and demonstrated in Table 4. The drug release followed zero-order, as determined by the comparison of computed values of the regression coefficient ($R^2$) for zero-order and first-order, implying anomalous transport as the most important mechanism of drug release. When the data was subjected to the Higuchi model it exhibited a molecular pattern of drug release, whereas in the Korsmeyer-Peppas model the diffusional exponent

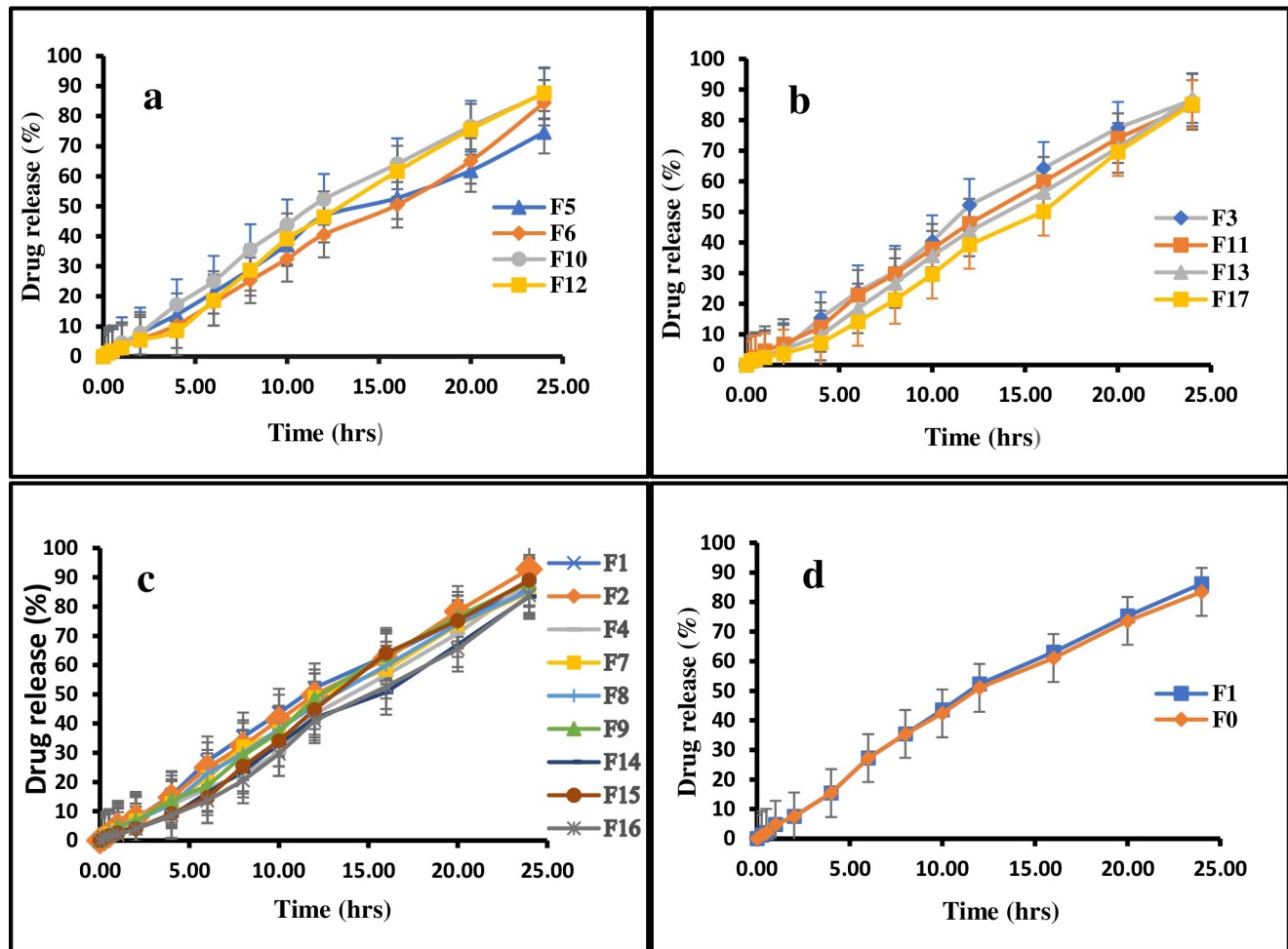

**Fig 4. Drug release profile of microparticles fabricated with HPMC = 200 mg (a), 800 mg (b), and 500 mg (c), and design proposed optimized formulation (F1) and developed optimized formulation (F0), (d).**

"n" values revealed a non-Fickian diffusion process from MLX loaded HPMC microparticles [37, 69].

**3.2.6 Optimization of formulation.** Design-Expert recommended the optimized formulation (F0) based on the percentage yield, percentage of EE, average particle size, and *in vitro* percent of cumulative drug release parameters. The predicted values in terms of percentage yield, percentage of EE, average particle size, and *in vitro* percent of cumulative drug release were 80.5%, 82.616%,160.526 μm, and 84.525%, respectively. The optimized formulation (F0) was successfully developed in triplicate using the design generated variables (polymer concentration: 500 mg, stirring speed: 1000 rpm, surfactant concentration: 1.0%). The developed optimized formulations (F0) were characterized for percentage yield, percentage of EE, average particle size, and *in vitro* percentage of cumulative drug release, its values were found to be 82.24 ±1.09%, 81.37 ±1.15%, 154.52 ±7.06 μm, and 83.43 ± 0.93%, respectively, as revealed in Table 5 and Fig 4(D). When the developed optimized formulations (F0) release data were subjected to the release kinetic modeling, it followed the non-Fickian mechanism (n = 0.843), and zero-order kinetic ($R^2$ = 0.9945), which is better than 1$^{st}$ order value ($R^2$ = 0.9863). The independent variable values incurred from the developed optimized formulations (F0) were

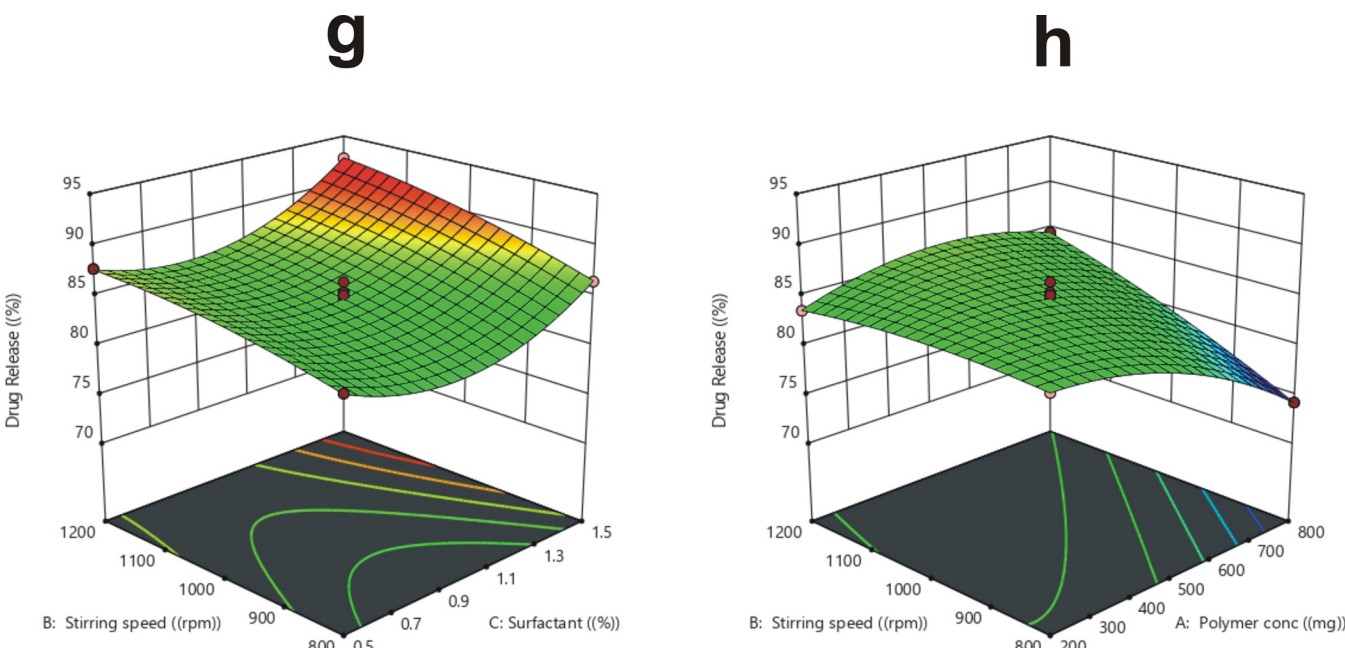

**Fig 5. The influence of stirring speed and surfactant concentration on *in vitro* drug release (g), the influence of stirring speed and polymer concentration on *in vitro* drug release (h).**

remarkably close to the design predicted values, which represents factual consistency, reliability, and validity of BBD in the colon targeted delivery of MLX loaded HPMC microparticles fabricated by the oil in oil (O/O)/ ESE method.

**Table 4. Kinetic modeling figures of *in vitro* drug release.**

| Code | Zero-Order | | First-Order | | Higuchi Model | | Korsmeyer Pappas Model | |
|---|---|---|---|---|---|---|---|---|
| | $R^2$ | $K_0$ | $R^2$ | $K_1$ | $R^2$ | $K_H$ | $R^2$ | $n$ |
| F1 | 0.9945 | 3.866 | 0.9822 | 0.062 | 0.9791 | 14.891 | 0.9851 | 0.757 |
| F2 | 0.9978 | 3.948 | 0.9550 | 0.062 | 0.8737 | 15.027 | 0.9921 | 0.822 |
| F3 | 0.9920 | 3.848 | 0.9668 | 0.060 | 0.8779 | 14.680 | 0.9737 | 0.736 |
| F4 | 0.9914 | 3.518 | 0.9310 | 0.051 | 0.8256 | 13.212 | 0.9763 | 0.768 |
| F5 | 0.9884 | 3.282 | 0.9840 | 0.048 | 0.9064 | 12.684 | 0.9840 | 0.779 |
| F6 | 0.9946 | 3.328 | 0.9464 | 0.046 | 0.8353 | 12.551 | 0.9974 | 0.847 |
| F7 | 0.9954 | 3.672 | 0.9682 | 0.056 | 0.8772 | 13.998 | 0.9863 | 0.804 |
| F8 | 0.9918 | 3.775 | 0.9806 | 0.059 | 0.9047 | 14.517 | 0.9784 | 0.820 |
| F9 | 0.9957 | 3.783 | 0.9512 | 0.057 | 0.8580 | 14.317 | 0.9857 | 0.753 |
| F10 | 0.9955 | 3.912 | 0.9795 | 0.063 | 0.9036 | 15.045 | 0.9906 | 0.794 |
| F11 | 0.9974 | 3.667 | 0.9663 | 0.055 | 0.8757 | 13.966 | 0.9880 | 0.695 |
| F12 | 0.9934 | 3.722 | 0.9463 | 0.056 | 0.8427 | 14.036 | 0.9941 | 0.783 |
| F13 | 0.9959 | 3.551 | 0.9454 | 0.052 | 0.8373 | 13.360 | 0.9780 | 0.831 |
| F14 | 0.9917 | 3.331 | 0.9409 | 0.047 | 0.8238 | 12.491 | 0.9661 | 0.637 |
| F15 | 0.9839 | 3.697 | 0.9166 | 0.054 | 0.8027 | 13.780 | 0.9719 | 0.720 |
| F16 | 0.9853 | 3.285 | 0.9265 | 0.046 | 0.9389 | 12.225 | 0.9861 | 0.763 |
| F17 | 0.9821 | 3.319 | 0.9159 | 0.046 | 0.8671 | 12.315 | 0.9775 | 0.835 |

**Table 5. The optimized formulation (F0) levels, predicted, and observed values.**

| Independent variables | | Optimized levels |
|---|---|---|
| Polymer concentration (mg) ($X_1$) | | 500 |
| Stirring speed (rpm) ($X_2$) | | 1000 |
| Surfactant concentration (%) ($X_3$) | | 1.00 |
| **Dependent variables** | **Predicated responses** | **Observed responses** |
| Percent yield (%) ($R_1$) | 80.5 | 82.24 |
| Entrapment efficiency (%) ($R_2$) | 82.616 | 81.37 |
| Particle size (μm) ($R_3$) | 160.526 | 154.52 |
| Drug release (%) ($R_4$) | 84.752 | 83.43 |

## 3.3. Fourier Transform Infrared Spectroscopic (FTIR) analysis

Fig 6(A) shows the FTIR spectrum of pure MLX, which shows a prominent peak at 3283.3130 cm$^{-1}$, which is thought to be due to secondary aliphatic amine (–R-N–H) stretching vibrations. The supposition is supported by a sharp peak at 1260.3027cm$^{-1}$, which is the result of aliphatic amine (–C–N) stretching vibration. The MLX spectra revealed a second strong peak at 1609.9024 cm$^{-1}$, indicating the presence of a secondary amide group (–CONH). In general, the bending vibration of amine (–N–H) follows the peak, secondary amide group, although these peaks are not visible in the MLX spectrum, this is due to the fact that MLX's–N–H group is a secondary aliphatic amine, which is normally weak and unnoticed. The peak at 1456.0386 cm$^{-}$

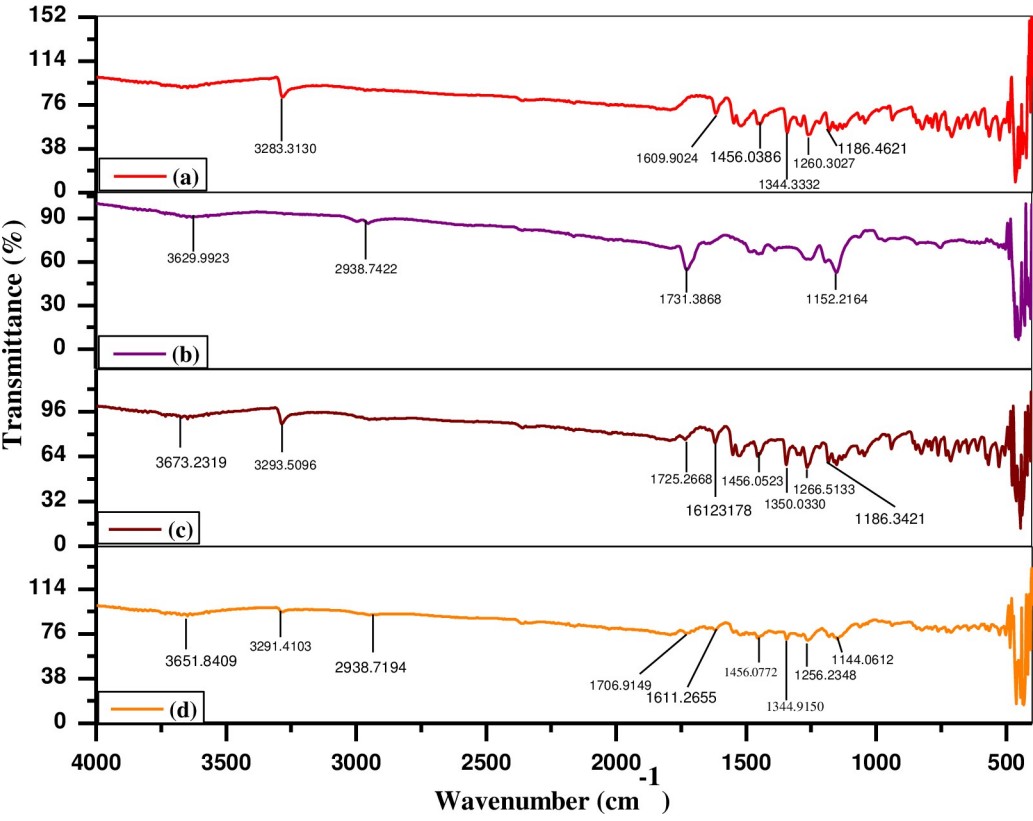

**Fig 6. FTIR spectra of MLX (a), HPMC (b), physical mixture of MLX and HPMC (c), as well as optimized formulation (F0) (d).**

[1], is related to a conjugated alkene group (–C = C–). The peaks found at 1344.3332 cm[-1] and 1187.8826 cm[-1], are linked with the S = O group, indicating asymmetric and symmetric stretching vibrations, respectively [15, 35, 70]. The presence of a peak at 3629.9923cm[-1] in the HPMC spectrum is due to the stretching of the O-H functional group [71]. A peak at 2939.7422 cm[-1] was observed owing to -C-H bond stretching [72]. The peak observed at 1731.3868 cm[-1] is related to C = C stretching and another peak found at 1152.2164 cm[-1] is attributed to stretching of the secondary alcoholic group, as revealed in Fig 6(**B**) [73]. The FTIR spectra of pure MLX and HPMC were compared to the FTIR spectra of the physical mixture, and it was found that MLX's characteristic peak had not changed significantly, as shown in Fig 6(C). The FTIR spectra of MLX loaded HPMC microparticles optimised formulation (F0) are shown in Fig 6(D), and revealed almost identical results to pure MLX with a miner shift in wave number. It was confirmed that all functional groups were in their respective ranges, indicating no drug-polymer interaction. These findings were in line with earlier research findings [35].

### 3.4. X-ray diffraction (XRD) analysis

The XRD patterns of the pure MLX, HPMC, physical mixture of MLX and HPMC, as well as optimized formulation (F0), are illustrated in Fig 7. The XRD pattern of pure MLX demonstrates individual intense peaks between the range of 6–30˚ at 2θ of diffraction angle

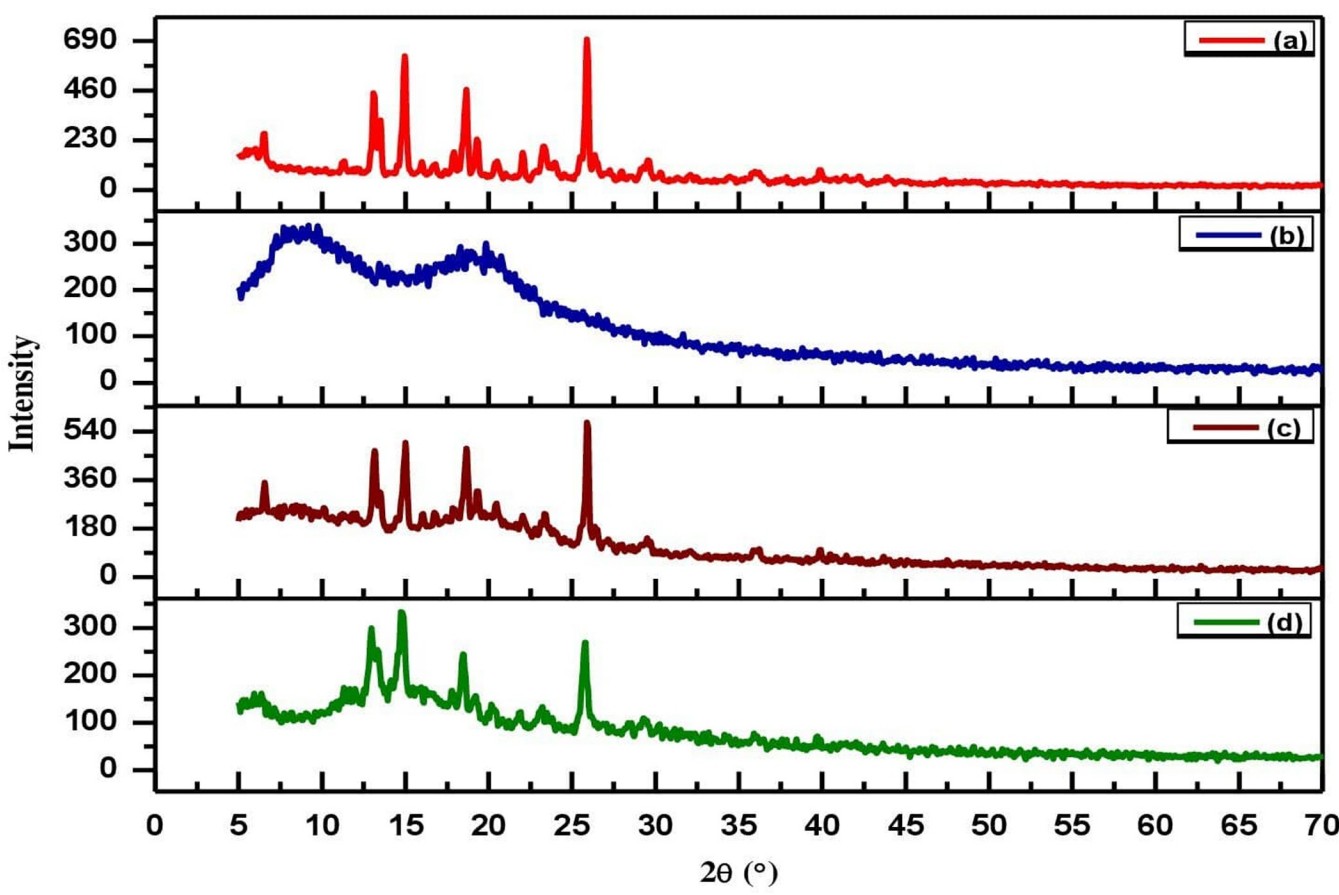

**Fig 7. XRD pattern of MLX (a), HPMC (b), physical mixture of MLX & HPMC (c), and optimized formulation (F0), (d).**

corresponding to 13.17˚, 15.06˚, 18.49˚, and 26.07˚, which demonstrates the crystalline nature of MLX, as shown in Fig 7(A) [38, 70]. The HPMC illustrated a wide hump in between the range of 5˚-25˚, which reflects its amorphous state, as displayed in Fig 7(B) [74]. The XRD pattern of the physical mixture of the MLX as well as HPMC, and optimized formulation (F0) maintained their peaks, which revealed that there was no interaction between the ingredients of the physical mixture and optimized formulation, as depicted in Fig 7(C) and 7(D) [75]. However, in the optimized formulation (F0) the lesser and wider peaks of MLX showed that the drug (MLX) was successfully encapsulated in the amorphous system of microparticles. This was due to the amorphous nature of created microparticles containing hydrophilic polymers, i.e., HPMC, which bestowed its characteristics to MLX. The development of microparticles that reduce the crystallinity of the MLX may help to improve the drug's solubility and dissolution [15].

## 3.5. Thermal stability via Differential Scanning Calorimetric (DSC) analysis

As shown in Fig 8, the thermodynamics of MLX, HPMC, physical mixture of MLX and HPMC, and optimization formulation (F0) were investigated using DSC to determine the thermal behavior of drug, excipients, and formulation. The MLX thermograph showed a

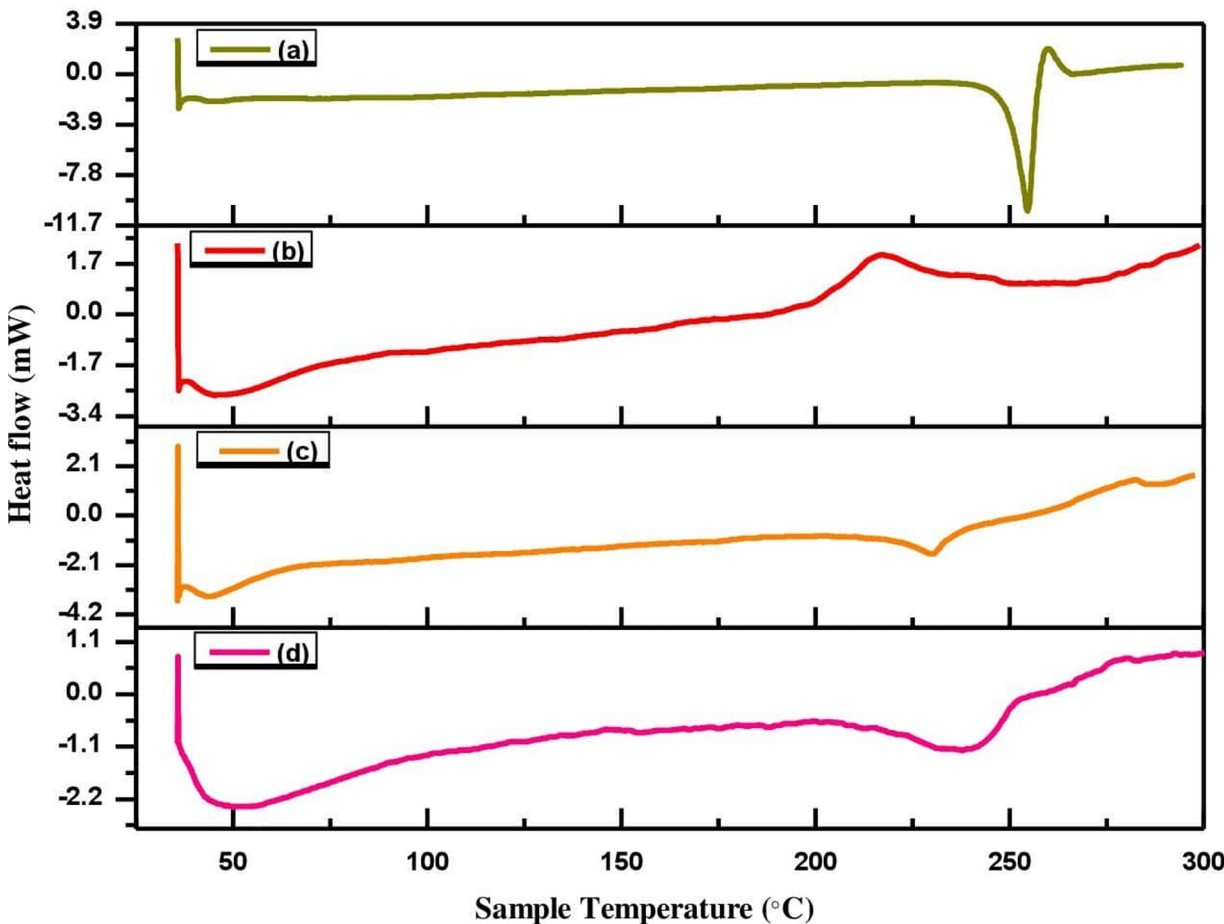

**Fig 8. DSC of MLX (a), HPMC (b), physical mixture of MLX as well as HPMC (c), and optimized formulation (F0), (d).**

typical sharp endothermic peak at 259.9 ˚C, which corresponds to its melting point, as shown in Fig 8(A) [76–78]. The MLX peak was suggested its crystalline nature [79]. In Fig 8(B), the HPMC thermograph showed endothermic and exothermic peaks at 53 ˚C and 218 ˚C, associated with the melting and decomposition temperature, respectively [80]. In the physical mixture of MLX and HPMC, the drug and polymer had retained their endothermic peaks, as shown in Fig 8(C) [81]. Moreover, in the DSC thermograph of optimized formulation (F0), as depicted in Fig 8(D), the shifting of MLX widened endothermic peak to the lower temperature 238 ˚C, suggesting that MLX was transformed from its crystalline state to the amorphous state [82].

### 3.6. Scanning Electron Microscopic (SEM) analysis

The surface morphology of the fabricated microparticles was examined at high resolution using scanning electron microscopy. The compact structure, smooth surface, and regularly spherical shape of the MLX loaded HPMC optimal formulation (F0) were validated by SEM images, which are attributed to the system's polymer concentration and stirring speed [37]. Fig 9(A)–9(C) shows SEM micrographs of MLX-loaded HPMC optimized formulation (F0), with smooth surface and spherical shape.

### 3.7. Acute oral toxicity study

An acute oral toxicity study was conducted to ascertain the toxicity and biocompatibility of the formulation in the rabbit's model and the OECD guidelines were followed. A simple and sensitive measure of adverse effects or signs of toxicity evolved with the consumption of hazardous chemicals/test formulations, resulting in body weight loss, essential organ atrophy, or both. All of group-I (control) measured parameters were compared to those of group-II (test) and any differences were studied in this study [47]. During the trial, there were no dead animals, no signs of disease, and no evidence of any seeable skin irritation/toxicity. On the 15th day, the rabbits were euthanized, and weighed the key organs, kept separately in 10% formalin solution for histopathological studies. The various parameters of hematological, biochemical, and weight variation analysis are reported in Table 6 and demonstrated insignificant changes in group-II (test) when compared with group-I (control) [83]. Histopathological studies of six vital organs, such as the heart, liver, spleen, stomach, lungs, and kidney, depicted no signs of abnormalities such as a lesion, disruption, hyperemia, and toxicity at the cellular level, as revealed in Fig 10. The absence of abnormalities shows the non-toxicity and biocompatibility of MLX loaded HPMC microparticles with the biological system of rabbits [84].

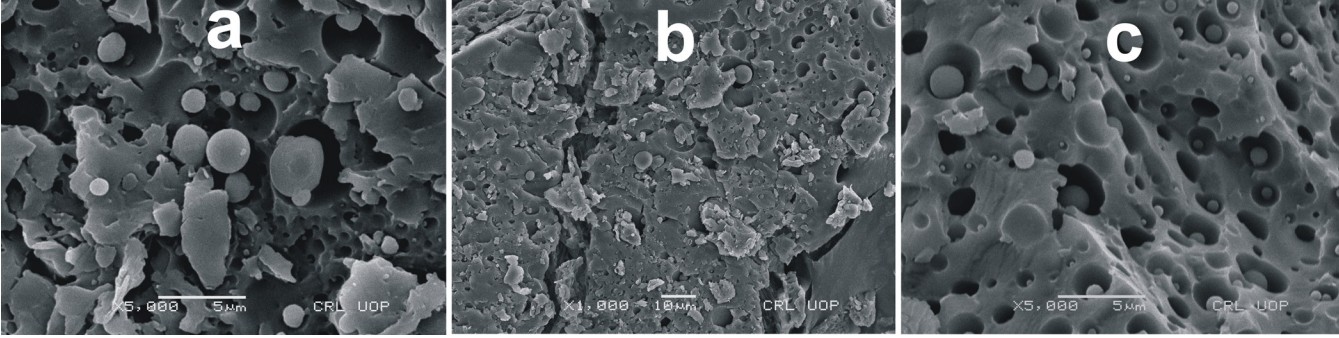

**Fig 9. SEM micrograph of F0 (a, b, & c).**

**Table 6. Hematological, biochemical, and weight variation analysis of group-I (control) and group-II (test).**

| Test/Parameters | Group-I (Control) | Group-II (test)) |
|---|---|---|
| | **I) Hematological Parameters** | |
| Hemoglobin (g/dl) | 12.94 ± 0.40 | 13.02 ±0.43 |
| RBCs (Red blood cells) $\times 10^6/mm^3$ | 6.14 ± 0.23 | 6.49 ± 0.58 |
| WBCs (White blood cells) $\times 10^9/l$ | 6.98 ± 0.48 | 07.08 ± 0.32 |
| Platelets $\times 10^9/l$ | 4.45 ± 0.46 | 4.27 ± 0.33 |
| Lymphocytes (%) | 61.63 ± 2.50 | 61.19 ± 2.89 |
| Monocytes (%) | 03.45 ± 0.37 | 03.16 ± 0.21 |
| Neutrophils (%) | 52.98 ± 3.52 | 54.64 ± 2.87 |
| Mean corpuscular volume (%) | 64.41 ± 2.06 | 66.61 ± 1.93 |
| Mean corpuscular hemoglobin (pg./cell) | 22.47 ± 0.69 | 23.38 ± 1.03 |
| Mean corpuscular hemoglobin conc (%) | 34.24 ±1.74 | 33.28 ± 1.11 |
| | **II) Biochemical Parameters** | |
| ALT/SGPT (IU/l) | 148.73 ± 3.32 | 157.95 ± 2.85 |
| AST/SGOT (IU/l) | 71.13 ± 3.21 | 67.32 ± 3.82 |
| Creatinine (mg/dl) | 1.19 ± 0.54 | 1.39 ± 0.32 |
| Serum uric acid (mg/dl) | 3.25 ± 0.63 | 3.51 ± 0.37 |
| Triglycerides (mg/dl) | 64.20 ± 4.08 | 66.73 ± 3.84 |
| Total cholesterol (mg/ dl) | 65.24 ± 3.43 | 63.59 ± 4.12 |
| Serum urea (mg/dl) | 17.02 ± 1.24 | 15.86 ± 1.72 |
| | **III) Weight (g)of Rabbit Organs** | |
| Heart | 4.33 ± 0.34 | 4.18 ± 0.23 |
| Liver | 7.80 ± 2.35 | 8.12 ± 2.17 |
| Spleen | 1.73 ± 0.54 | 1.61 ± 1.03 |
| Kidney | 11.16 ± 1.79 | 10.49 ± 1.41 |
| Stomach | 12.42 ± 1.32 | 12.92 ± 0.96 |
| Lung | 9.94 ± 0.57 | 9.94 ± 0.57 |

All values are described in mean ±SD (n = 3).

## 3.8. Conclusion

The oil in oil (O/O)/ ESE technique was used to successfully fabricate colon targeted MLX loaded HPMC microparticles in this study. Based on percentage yield, percent EE, average particle size, and in vitro percentage of cumulative drug release, all formulations were created and optimized using design expert software. Physicochemical characterization of formulations by FTIR, XRD, and DSC analysis showed that all components of the formulation were compatible. SEM images depicted a compact structure with smooth surface and spherical shape microparticles. MLX was decently encapsulated in an amorphous state with maximum EE. The *in vitro* cumulative drug analysis revealed that the MLX loaded HPMC microparticles release the drug in a decelerated manner in gastric milieu for the first 2–3 hrs, followed by a controlled release for 24 hrs, with zero-order release kinetics and non-Fickian mechanism, which is the goal of the colon targeted drug delivery system. The MLX loaded HPMC optimized formulation (F0) was successfully developed and evaluated for independent variables values. The obtained values were very close to the design predicted values, indicating the consistency and reliability of BBD. An acute oral toxicity study confirmed the non-toxicity and biocompatibility of MLX-loaded HPMC microparticles with the biological system. Consequently, it is expected that MLX-loaded HPMC microparticles, particularly the optimized formulation (F0),

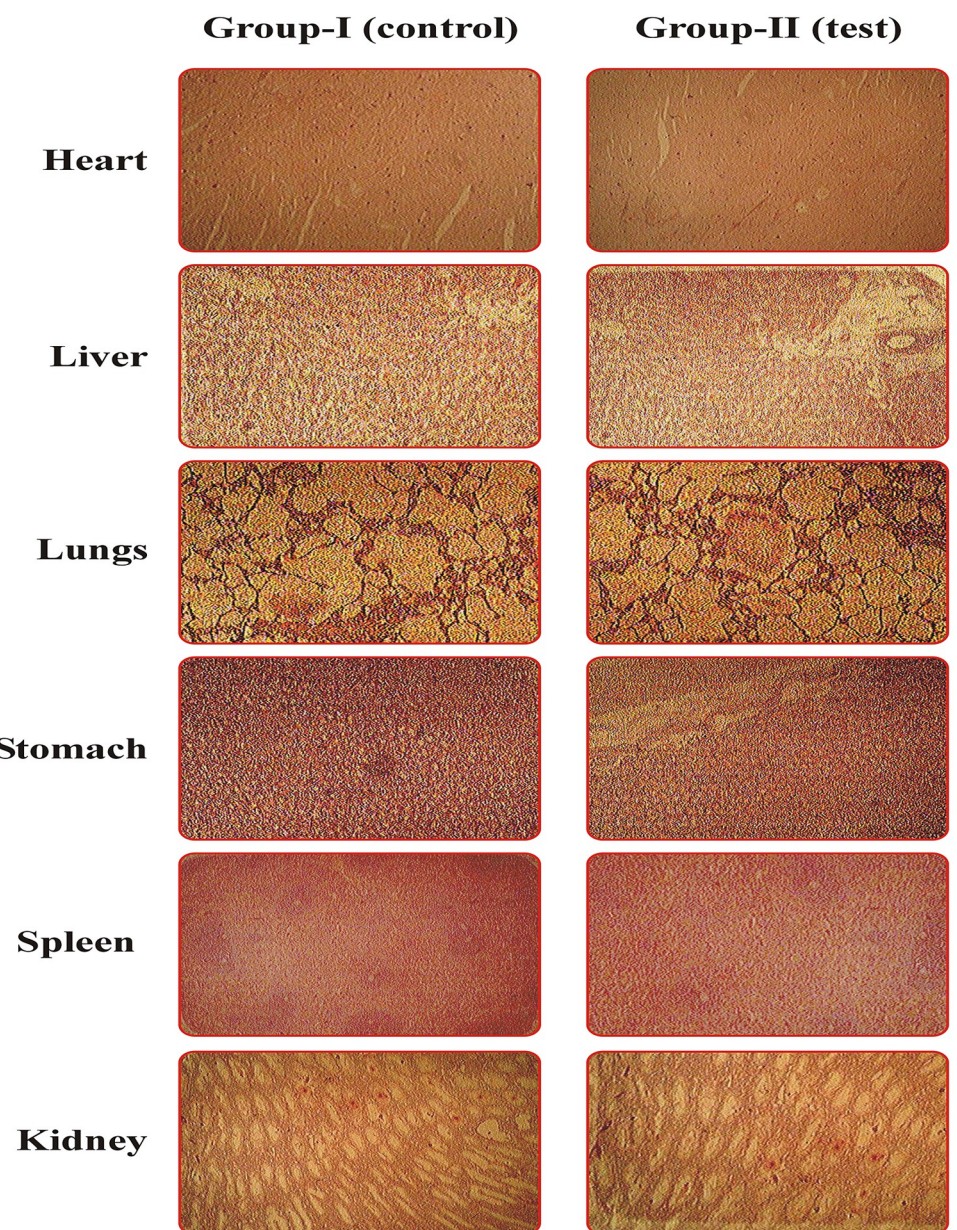

**Fig 10. Histopathological microscopic examinations of vital organs of rabbit of Group-I (control) and Group-II (test).**

can be used as a viable alternative for the treatment of CRC by safe and controlled management with the best patient compliance.

## Acknowledgments

The authors of this study would like to acknowledge the Faculty of Pharmacy at, The Islamia University Bahawalpur (Pakistan) for furnishing research facilities and a cordial environment for accomplishing this work. The authors are also thankful to English Pharmaceuticals Industries, Lahore (Pakistan) and Martin Dow Marker limited (formerly MERCK Pvt Ltd), Quetta (Pakistan), for supplying Meloxicam and HPMC, respectively to carry out this study.

## Author Contributions

**Conceptualization:** Syed Abdul Wasay, Muhammad Akhtar.

**Data curation:** Sobia Noreen.

**Methodology:** Syed Abdul Wasay.

**Resources:** Muhammad Akhtar.

**Software:** Muhammad Akhtar.

**Supervision:** Syed Umer Jan.

**Validation:** Muhammad Akhtar.

**Visualization:** Rahman Gul.

**Writing – review & editing:** Sobia Noreen, Rahman Gul.

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
