## [Decision Letter · Decision Letter 0]

26 Dec 2021

PONE-D-21-36281Meloxicam loaded microparticles for colon targeted delivery: statistical optimization, physicochemical characterization, and in-vivo toxicity studyPLOS ONE

Dear Dr. Umer Jan,

Thank you for submitting your manuscript to PLOS ONE. After careful consideration, we feel that it has merit but does not fully meet PLOS ONE’s publication criteria as it currently stands. Therefore, we invite you to submit a revised version of the manuscript that addresses the points raised during the review process.

We look forward to receiving your revised manuscript.

Kind regards,

Vineet Kumar Rai, PhD

Academic Editor

PLOS ONE

Journal Requirements:

2. To comply with PLOS ONE submissions requirements, in your Methods section, please provide additional information regarding the experiments involving animals and ensure you have included details on (1) methods of sacrifice, and (2) efforts to alleviate suffering.

4. PLOS requires an ORCID iD for the corresponding author in Editorial Manager on papers submitted after December 6th, 2016. Please ensure that you have an ORCID iD and that it is validated in Editorial Manager. To do this, go to ‘Update my Information’ (in the upper left-hand corner of the main menu), and click on the Fetch/Validate link next to the ORCID field. This will take you to the ORCID site and allow you to create a new iD or authenticate a pre-existing iD in Editorial Manager. Please see the following video for instructions on linking an ORCID iD to your Editorial Manager account: https://www.youtube.com/watch?v=_xcclfuvtxQ.

Reviewers' comments:

Reviewer's Responses to Questions

**Comments to the Author**

1. Is the manuscript technically sound, and do the data support the conclusions?

Reviewer #1: Yes

Reviewer #2: Partly

2. Has the statistical analysis been performed appropriately and rigorously? 

Reviewer #1: Yes

Reviewer #2: Yes

3. Have the authors made all data underlying the findings in their manuscript fully available?

Reviewer #1: Yes

Reviewer #2: Yes

4. Is the manuscript presented in an intelligible fashion and written in standard English?

Reviewer #1: Yes

Reviewer #2: No

5. Review Comments to the Author

Reviewer #1: Dear Authors,

This is an interesting study and the authors have collected a unique dataset using cutting edge methodology. The paper is generally well structured. However, in my opinion the paper has some shortcomings in regards to some text, and I feel this dataset has not been utilized to its full extent. Below I have provided numerous remarks on the text.

1- In line (1), Let the title tells what you did on Meloxicam loaded microparticle. I suggest “Developed meloxicam loaded microparticles for colon targeted delivery: statistical optimization, physicochemical characterization, and in-vivo toxicity study”.

2- The complete name of the HPMC should be written in the first time you stated.

3- In line (27), replace “predicted” to “confirmed”.

4- In the introduction sector, what is the problem which the study try to solve or what is the objective of the work.

5- In the introduction sector, authors should include literature review on the topic of the manuscript. microparticles and HPMC microparticles.

6- In materials and methods sector, what is the base you depend on selecting the value of the independent variables.

7- The included figures (figures 5, 6, 7, 9, and 10) are not clear). Authors should add images for suitable resolution.

8- The magnification power of figure 4 should be included and clear for all the microparticles images.

9- Figure 4 indicates that there is microparticles agglomeration.

10- In line (56), replace “drug” with “MLX” in the equation used for the Entrapment Efficiency determination.

11- In lines (110, 111), is this (1) related to references or the number of the equation. I guess it is relation the equation so it should be differentiated from the format used with references.

12- In lines 111, 140, and 150, 167 207 the same notice as before.

13- In line (194), write the complete letters of the word “eq”.

14- In line (220), write little details about the gastric gavage, including needle used.

15- In line (235), write “the optimized MLX loaded microparticles” instead of “microparticles” alone.

16- In line 244, In the results and discussion section, write some introductory sentence including the aim of the study and that you prepared the microparticles.

17- The discussion lakes relation with the literature review and more mechanistic discussion is required.

18- In lines (449 and 450), include the name of the obtained microparticles and the name of the animal used in the experiment.

19- Generally, more clarification is required in terms of the language.

Reviewer #2: In this study authors’ developed meloxicam (MLX) loaded HPMC microparticles for colon targeted delivery to treat colorectal cancer (CRC). The physicochemical characterizations of the developed formulations were conducted using the suitable methods and the toxicity study was performed using animal model. This work is well-designed but it has significant lacking in methodology, data presentation and overall writing. Although different organs have been examined to check the toxicity/biocompatibility of the formulation, authors’ did not conducted the study to see how much drug is going to the colorectal area which is the main target for this development. The manuscript is poorly written and the “Results and discussion” section doesn’t have the flow/coherency. Moreover novelty/importance of this study has not been clearly mentioned in the manuscript. Following issues need to be addressed.

General comments:

-English needs to be improved

-Typos need to be corrected

-Inconsistencies in using terminologies (mentioned in specific comments), references need to be addressed.

Specific comments:

Abstract

Page 1, line 12, Need to add e after “Th”

Page 1, line 21-Need a full stop after performed

Page 1, lines 23-25, since the results were mentioned as range, don’t need the standard deviations and start from low to high values. e.g. the percentage yield is 65.75-90.71% etc.

Introduction

In the first para need to mention what are the current treatments for CRC then how MLX is related to the CRC treatment.

In the 2nd para mention everything about MLX. E.g. chemistry, what study (I mean formulation work) has been done on MLX related to CRC treatment, limitations of the existing research on MLX etc.

Currently the importance of this work is not clearly mentioned, so bring that discussion in the introduction to show the importance/novelty of this work.

Page 3, line 59, what do you mean by controlled liberation, is it controlled release? If so mention controlled release otherwise it is confusing the readers. Throughout the manuscript mention controlled release.

Page 3, line 78, I think the word should be statistical not statical

Page 3, lines 80-81, this is an incomplete sentence. Mention targeting of what…

Materials:

Please keep consistency in mentioning the manufacturer’s name. Currently for some you mentioned city and country and others are only country.

Page 4, line 91, need a comma before Quetta

Page 4, line 93, (England, UK), is England a city here?

Page 4, line 94, delete comma before (Germany)

Method:

Page 4, lines 99-100, this sentence doesn’t make any sense. Please rewrite this. In addition, in this sentence it should be statistical tool not statical tool

Page 4, lines 100-101, If I don’t do any mistake, for 3-level and 3-factor design total number of formulations should be 27. Why only seventeen formulations were prepared?

Please keep consistency in mentioning the terminologies, either use variables or parameters but not mixed-up of both which is confusing.

Page 4, lines 108-109, “….practicing State-Ease…” Please clearly mention in the sentence what is State-Ease?

Page 5, Table 1, the unit for Mean particle size should be µm not µg.

Page 7, line 159, please mention what type of particle size was measured, like volumetric, geometric etc.

Page 8, section 2.3.5, need to re-write this section in a good English. Some information are repetitive and some sentences are incorrect. How the optimized formulation (F0) came here? There is no information about F0 in the Table 1.

Page 8, section 2.3.6, please add the information about scanning speed and step for XRD runs. It is not clear what did you mean by measured amount, please mention approximate amount (e.g. 5-10 mg). Need to re-write this section in a good and correct English.

Page 8, section 2.3.7, please mention the different resolutions/magnifications used to collect the images. What did you mean by optimized quantity, did you try different amount of powder to make SEM samples and found out the optimized quantity? If not just mention powders samples were mounted……Also “….the stub was covered with gold using SEM” is not correct. I imagine the stub was coated with gold using as specific coater, please mention correctly with the name of the coater.

Page 9, section 2.3.8, It is not clear why in different phases, different mediums were used for different times. Line 205, what is configuration curve, did you mean calibration curve, if so mention calibration curve. Again delete drug liberation, instead use drug release throughout the manuscript.

Page 9, section 4-The numbering of this section and following sections is wrong. Please correct it.

Results and discussion:

Page 10, line 255-correct the word peek to peak

Page 11, line 260, No prior information mentioned, how the optimized formulation was obtained. What happened to all other formulations mentioned in Table 1?

Page 11, line 260-261, this statement may not be correct. Why some peaks are absent in F0 which were present in the physical mixture and even in the MLX and HPMC, need explanation. Does it mean interaction?

Page 11, lines 271-275, The reduction in the intensity of some peaks could be due to the difference in the ratio of MLX and HPLC in the physical mixture and F0 than the only MLX and HPMC. However, there is an absence of the first peak (at around 5 2theta position) in the F0 which is present in physical mixture as well as MLX diffractograms. Does it mean interaction? Please discuss correctly.

Page 12, lines 286-291, Again, both in the physical mixture and F0, there is a sign of interaction? Please justify this findings with other solid-state characterizations (XRD and FTIR).

Page 12, lines 295-296, “…..mainly depends on the polymer concentration and stirring speed of the system” How do we know that since no data on different speed and concentration has been shown here. Why suddenly F1 came in this study while for XRD, FTIR, and DSC, only F0 has been shown.

Page 12, lines 302, the range should be mentioned from low to high, e.g. the percentage yield (R1) ranged from 65.75-91.71%%. Figures 5-7 and 9 are not clear. Please replace with high resolution and clear images.

Page 19, Table 5, delete the units from the column “Observed responses” as units are already mentioned in the first column “Dependent variables”

Pages 12 to 19- From “Box Behnken Design to “Optimization of formulation” should be moved before all other solid-state characterizations, I mean before FTIR, XRD, DSC etc.

Table 6, please mention the unit (e.g. mg/g etc.) for the “Weight of Rabbit Organs”

Conclusion:

Page 21, lines 457-458, “Physicochemical characterization of formulations by FTIR, XRD, and DSC analysis showed that all components of the formulation are compatible”-this is not correct, since at least in DSC and FTIR there is an evidence of interaction. Please state whatever is in the data, don’t overstate.

References:

Inconsistency, some has journal names and some are missing. Please keep it consistent according to journal guidelines.

6. PLOS authors have the option to publish the peer review history of their article (what does this mean?). If published, this will include your full peer review and any attached files.

Reviewer #1: **Yes: **Tarek M. Faris

Reviewer #2: No

---

## [Author Response · Author response to Decision Letter 0]

15 Feb 2022

Reviewer 1: We are thankful to honorable reviewers 1 for their valuable time and useful contribution, as their inputs have helped us to improve the manuscript. 

Reviewer 2: We are also thankful to honorable reviewers 2 for their valuable time and useful contribution, as their inputs have helped us to improve the manuscript.

---

## [Editor Report · Decision Letter 1]

6 Apr 2022

Developed meloxicam loaded microparticles for colon targeted delivery: Statistical optimization, physicochemical characterization, and in-vivo toxicity study

PONE-D-21-36281R1

Dear Dr. Jan,

We’re pleased to inform you that your manuscript has been judged scientifically suitable for publication and will be formally accepted for publication once it meets all outstanding technical requirements.

Kind regards,

Vineet Kumar Rai, PhD

Academic Editor

PLOS ONE
---

## [Editor Report · Acceptance letter]

11 Apr 2022

PONE-D-21-36281R1 

Developed meloxicam loaded microparticles for colon targeted delivery: Statistical optimization, physicochemical characterization, and *in-vivo* toxicity study 

Dear Dr. Jan:

I'm pleased to inform you that your manuscript has been deemed suitable for publication in PLOS ONE. Congratulations! Your manuscript is now with our production department. 

Kind regards, 

on behalf of

Dr. Vineet Kumar Rai 

Academic Editor

PLOS ONE